# m^6^A mRNA Modifications in Glioblastoma: Emerging Prognostic Biomarkers and Therapeutic Targets

**DOI:** 10.3390/cancers16040727

**Published:** 2024-02-09

**Authors:** Gloria S. Xie, Hope T. Richard

**Affiliations:** 1Martel College, Rice University, Houston, TX 77005, USA; gsx1@rice.edu; 2Department of Pathology, School of Medicine, Virginia Commonwealth University, Richmond, VA 23219, USA; 3VCU Massey Cancer Center, Virginia Commonwealth University, Richmond, VA 23219, USA

**Keywords:** N6-methyladenine mRNA modification, glioma, prognostic biomarkers, therapeutic targets

## Abstract

**Simple Summary:**

N6-methyladenosine is a prominent epigenetic modification identified on mRNA, playing a pivotal role in shaping RNA function and emerging at the forefront of cancer research. Recent studies highlight the abnormal activation of m^6^A modification in glioma, demonstrating its crucial involvement in diverse aspects of glioma tumorigenesis. This review provides an overview of significant advancements in this field, particularly focusing on the downstream functional effects of m^6^A modification, the mechanisms underlying the dysregulation of m^6^A-related genes, and the therapeutic potential and roles of m^6^A modulators in treatment resistance of glioma.

**Abstract:**

Glioblastoma, the most common and aggressive primary brain tumor, is highly invasive and neurologically destructive. The mean survival for glioblastoma patients is approximately 15 months and there is no effective therapy to significantly increase survival times to date. The development of effective therapy including mechanism-based therapies is urgently needed. At a molecular biology level, N6-methyladenine (m^6^A) mRNA modification is the most abundant posttranscriptional RNA modification in mammals. Recent studies have shown that m^6^A mRNA modifications affect cell survival, cell proliferation, invasion, and immune evasion of glioblastoma. In addition, m^6^A mRNA modifications are critical for glioblastoma stem cells, which could initiate the tumor and lead to therapy resistance. These findings implicate the function of m^6^A mRNA modification in tumorigenesis and progression, implicating its value in prognosis and therapies of human glioblastoma. This review focuses on the potential clinical significance of m^6^A mRNA modifications in prognostic and therapeutics of glioblastoma. With the identification of small-molecule compounds that activate or inhibit components of m^6^A mRNA modifications, a promising novel approach for glioblastoma therapy is emerging.

## 1. Introduction

Gliomas are the most commonly diagnosed malignant CNS tumors in the United States, accounting for 80% of adult malignant brain tumors in the country [1,2,3]. Among gliomas, glioblastoma (CNS WHO grade 4 tumor) has the worst prognosis. More than 12,900 cases of glioblastoma are diagnosed each year [1,2,3]. The frequency of glioblastoma is higher in male patients than in female patients, with 5.6 cases per 100,000 population compared to 3.5 cases [2]. Treatment protocols for newly diagnosed glioblastoma include surgical resection, followed by radiotherapy and chemotherapy with temozolomide (TMZ). Unfortunately, surgical resection of glioblastomas is often incomplete, and they exhibit resistance to radiotherapy and chemotherapy [1,2,3]. The remaining tumor cells rapidly grow and invade nearby normal brain tissues, leading to brain damage and patient death. As a result, glioblastoma has an aggressive disease course, with a median survival of 15 months [1,2,3], and the five-year survival rate for glioblastoma patients is only 6.8 percent [2].

mRNA’s functions, such as transcription, are not only determined by their nucleotide sequences but also by chemical modifications on the mRNAs, known as the epitranscriptome. Among all modifications, N6-methyladenosine (m^6^A) formation is a high-frequency modification on RNAs in mammalian cells [4,5,6,7]. M^6^A is generated by a “writer” complex, which contains methyltransferase-like 3 (METTL3), methyltransferase-like 14 (METTL14), WT1-associated protein (WTAP), vir-like m^6^A methyltransferase associated (VIRMA), and RNA-binding motif protein 15 (RBM15). In the m^6^A write complex, METTL3 and METTL14 are the catalytic components [6,7]. M^6^A sites in RNA are recognized by “readers”, such as YTH N6-methyladenosine RNA-binding protein [YTHDF] and YTH-domain-containing (YTHDC) family members. The binding of the readers to their target RNAs dictates the downstream effects of m^6^A modification. As a result, the m^6^A modification process regulates the fate of mRNA including mRNA stability, mRNA translation, mRNA splicing, and nuclear export of mRNA [6,7], thereby leading to increases or decreases in gene expression. M^6^A methylation may be reversible since m^6^A can be demethylated by “erasers”. At present, two m^6^A erasers have been identified, which are Fat mass and obesity-associated protein (FTO) and AlkB homolog 5 RNA demethylase (ALKBH5) [6,7]. Intriguingly, the biological roles of m^6^A RNA modification have been indicated in development, cellular differentiation, and response to environmental stimuli, etc. Moreover, the dysregulation of m^6^A mRNA modification and related proteins has been implicated in a in disease states, such as cancer, neurodegenerative disorders, and type 2 diabetes [8,9,10], underscoring its significance in both normal physiological functions and pathological conditions.

To investigate the significance of m^6^A, it is crucial to accurately identify m6A sites through mapping and quantitative analysis within cells. The techniques employed for profiling m^6^A modifications have undergone rapid development in recent years. The initial discovery of N6-Methyladenosine dates back to 1969, using 2D-TLC (thin-layer chromatography) [11]. Subsequently, the SCARLET method (site-specific cleavage and radioactive-labeling followed by ligation-assisted extraction and TLC) emerged as a technique to quantify m^6^A levels at a specific position in RNA from cells [12]. In addition, researchers employed RNA immunoblot assays with an anti-m^6^A antibody, including dot-blot and immune-Northern blot, to identify global changes in m^6^A in RNA [13]. LC-MS/MS represents an alternative approach for detecting and measuring m^6^A modified RNA. The integration of HPLC with MS/MS has further advanced this method, enabling elevated sensitivities [14]. A breakthrough came in 2012 with the introduction of the m^6^A-seq or MeRIP-seq technique, enabling transcriptome-wide mapping of m^6^A [5,15]. This approach relies on antibody-based enrichment of m^6^A-containing RNA fragments, coupled with high-throughput sequencing, providing accurate and quantitative insights into m^6^A distribution across the entire transcriptome. In tandem with this method, MeRIP-qRT-PCR utilizes quantitative reverse transcription polymerase chain reaction (qRT-PCR) to assess modified RNA fragments enriched by m^6^A, offering insights into m^6^A loci within single genes. More recently, to enhance the sequencing resolution, modified versions of MeRIP-seq/m^6^A-seq have emerged, employing cross-linking techniques to locate m^6^A sites with greater precision. Examples include miCLIP and m^6^ACLIP. These cross-linking immunoprecipitation (CLIP) methods achieve single-nucleotide resolution, which has led to their increased popularity [16]. Several m^6^A sequencing methods that do not rely on antibodies have been developed, including m^6^A-REF-seq and DART-seq. In m^6^A-REF-seq, a methylation-sensitive RNA endonuclease MazF is utilized to cleave unmethylated ACA motifs, thereby enhancing the sensitivity of detecting m^6^A sites [17]. On the other hand, DART-seq employs the m^6^A-binding domain YTH for m^6^A recognition. Notably, DART-seq is capable of identifying m^6^A residues in cellular RNAs using minimal RNA quantities [18]. Furthermore, DART-Seq can be seamlessly integrated into standard RNA-seq library preparation methods.

## 2. m^6^A RNA Modification and Associate Proteins as Prognostic Biomarkers in Glioma

To date, significant efforts have been dedicated to large-scale human data mining focused on m^6^A RNA methylation and its associated proteins for the identification of prognostic biomarkers in glioma. This extensive data mining entails the extraction and analysis of substantial datasets related to the presence and impact of m^6^A RNA modifications in individuals with glioma. The process involves extracting pertinent information from datasets, encompassing m^6^A RNA expression levels, clinical data, and patient outcomes. Primary sources for these datasets include reputable repositories such as The Cancer Genome Atlas (TCGA), Gene Expression Omnibus (GEO), and other relevant databases. Individual m^6^A RNA methylation regulators, including METTL3, ALKBH5, YTHDF2, and IGF2BP3, have been assessed in patient datasets to examine their expression levels across normal tissues, lower-grade gliomas, and glioblastomas, as well as their prognostic significance [19,20,21,22,23,24,25,26]. In gliomas, the expression levels of METTL3, ALKBH5, YTHDF2, and IGF2BP3 were found to be elevated compared to normal brain tissues [19,20,21,22,23,24,25,26]. Furthermore, the individual expression level of each protein is indicative of a poor prognosis in malignant glioma patients [19,20,21,22,23,24,25,26].

Comprehensive analyses of a cluster of m^6^A RNA methylation regulators in glioma using patient databases has also been documented from several studies. Recently, the correlation between multiple m^6^A methylation regulators in glioma and normal samples using a TCGA dataset has been explored [25]. The majority of m^6^A RNA methylation regulators exhibited distinct expression patterns in gliomas compared to normal tissues. The relationships between m^6^A methylation regulators and gliomas were further examined using the Gene Expression Profiling Interactive Analysis (GEPIA) tool. The results showed that METTL14, FTO, YTHDF1, and YTHDF3 demonstrated significant upregulation in low-grade gliomas compared to normal tissues. In addition, elevated levels were observed for RBM15B, WTAP, FTO, YTHDF2, YTHDF3, IGF2BP2, and IGF2BP3 in glioblastoma samples compared with normal brain controls. Next, the study investigated the correlation between m^6^A-related genes and the prognosis of gliomas by evaluating the expression levels of these genes and the survival outcomes of glioma patients. Univariate analysis revealed that high expression of METTL14, RBM15, ZC3H13, WTAP, FTO, ALKBH5, YTHDF1, YTHDF2, YTHDF3, YTHDC1, YTHDC2, IGF2BP2, and IGF2BP3 was associated with an unfavorable prognosis for gliomas. Furthermore, multivariate Cox regression analysis indicated that high expression of YTHDC2 served as an independent positive prognostic factor for overall survival, while elevated expression of IGF2BP3 acted as an independent negative prognostic factor for the overall survival of glioma patients [26].

The value of a high IGF2BP3 level as an indicator predicting poor prognosis in glioma patients was confirmed by another study. This study conducted an analysis of the variations in gene expression of m^6^A RNA methylation modulators between low-grade and high-grade gliomas using TCGA data [26]. The examination of 19 m^6^A RNA methylation modulators in gliomas highlighted IGF2BP3 as the most markedly altered gene associated with m^6^A RNA methylation. Subsequent analysis demonstrated that patients with gliomas exhibiting high IGF2BP3 expression experienced a significantly diminished probability of survival compared to those with low IGF2BP3 expression.

A study analyzing microarray data from 605 glioma cases in TCGA database confirmed YTFDF1’s significance as a predictor of poor prognosis [27]. Within the m6A RNA methylation modulators, YTHDF1 emerged as a negative prognostic indicator, while RBM15 and METTL14 were identified as positive indicators for patient prognosis. Notably, the microRNA hsa-mir-346 exhibits the ability to bind to the 3′UTR of YTHDF1, leading to the negative regulation of YTHDF1 expression in glioma cells [27]. This regulatory interaction may be associated with mRNA stability, as microRNAs typically induce a decrease in mRNA stability. This research revealed the involvement of microRNA in m^6^A methylation through the regulation of m^6^A modulator expression. Remarkably, the interaction between microRNA and m^6^A methylation is bidirectional, indicating that m^6^A methylation is a significant post-transcriptional modification for microRNA as well. For example, METTL3 plays a crucial role in the fundamental expression of the majority of miRNAs by augmenting miRNA maturation in both cancerous and non-cancerous cell lines [28]. The microRNA biogenesis process includes the processing of primary microRNAs by the microprocessor complex, where the RNA binding protein DGCR8 serves as a major component. METTL3 can methylate pri-miRNAs, effectively marking them for binding and processing by DGCR8, consequently facilitating the maturation of miRNAs [28].

Furthermore, an m^6^A regulatory prognostic signature has been established by leveraging a comprehensive analysis of m^6^A regulatory genes, utilizing multi-omics data sourced from glioma patients in TCGA and normal brain tissues in the Genotype-Tissue Expression (GTEx) database [29]. R package iClusterPlus software (http://www.bioconductor.org/packages/devel/bioc/html/iClusterPlus.html, accessed on 5 February 2024) was employed to analyze the multi-omics data, encompassing the expression of m^6^A regulatory genes. The risk signature model was formulated through univariate and multivariate Cox analyses of the expression levels of m^6^A regulatory genes. The Kaplan–Meier method was applied to evaluate the overall survival disparity between high- and low-expression groups. Notably, the risk signature comprises eight specific genes: ALKBH5, HNRNPA2B1, IGF2BP2, IGF2BP3, RBM15, WTAP, YTHDF1, and YTHDF2 [29]. The presence of this m^6^A regulatory signature is associated with an adverse prognosis in glioma patients, as evidenced by the TCGA data. Moreover, validation of this risk signature for glioma patient survival was performed using related clinical data from the Chinese Glioma Genome Atlas (CGGA) datasets [29]. These findings imply that the m^6^A regulatory gene signature can function as a molecular marker for predicting the prognosis of glioma.

## 3. m^6^A RNA Methylation in Gliomas: Functions and Downstream Target Genes

Numerous downstream genes of m^6^A RNA modification have been identified. Altered expression and regulation of those genes have been shown to play important roles in various aspects of cancer biology including cell proliferation and cell cycle, cancer cell stemness, DNA damage and repair, cell apoptosis and death, cell migration and invasion, and immunoregulation [6,7,8,9]. Thus, the effects of m^6^A mRNA modification on cancer biology of glioma are mediated, at least in part, through the expression of downstream target genes [10]. Moreover, the impact of m^6^A mRNA modification on biological processes and phenotypes of cancers does not rely on the global level of cancer cells. In fact, METTL3 and ALKBH5 are supposed to have opposite roles in regulating cell proliferation of glioma because METTL3 increases m^6^A level but ALKBH5 reduces m^6^A levels. However, METTL3 and ALKBH5 were both found to maintain the growth of glioblastoma cells [19,20,21]. Therefore, the impact of m^6^A mRNA modification on biological processes and phenotypes of cancers likely depend on the specific changes in the methylation state of genes caused by the changes in the individual m^6^A writer, eraser and/or reader.

### 3.1. Regulating Cell Proliferation and Cell Cycle Progression

Understanding the detailed mechanisms of regulation in the cell cycle is crucial for developing insights into diseases like cancer, where dysregulation of the cell cycle is a common feature. Recent studies have indicated that ALKBH5 and YTHDF2 are important regulators of glioma cell proliferation.

Zhang et al. found that high ALKBH5 expression in gliomas is correlated with shorter survival of patients, while low ALKBH5 expression is associated with longer survival of patients [21]. Therefore, ALKBH5 overexpression is indicative of an unfavorable prognosis in glioma patients. The function of ALKBH5 in gliomas was linked to cell proliferation and cell cycle progression [21]. ALKBH5 knockdown on glioblastoma-derived stem-like cells (GSCs) was shown to reduce their proliferation. Additionally, the knockdown of ALKBH5 in GSCs led to an elevation in the proportions of cells in the G0/G1 phase, coupled with a decrease in the proportions of cells in the S and G2/M phases of the cell cycle. Furthermore, knockdown of ALKBH5 impaired the tumorigenicity of GSC cells because the brain tumor formation rate of the GSCs with ALKBH5 knockdown was smaller as compared with control cells. The functions of ALKBH5 in cell proliferation and cell cycle progression could be attributed to its downstream target gene FOXM1. FOXM1 is a master cell cycle transcription factor for the transition from G1 to S phase as well as progression to mitosis. FOXM1 has been shown to regulate transcription of cell cycle genes essential for G1/S and G2/M progression and chromosome stability and segregation, such as cyclin D1 [30]. Regarding glioma, FOXM1 is one of the most frequent molecular alterations in the malignancy [31]. FoxM1 contributed to glioma progression and malignancy by being involved in cell proliferation, angiogenesis, invasion and maintenance of cancer cell stemness [31,32,33,34]. ALKBH5’s demethylation activity specifically targets newly synthesized transcripts of FOXM1 [21]. Demethylation of the transcripts by ALKBH5 enhanced the expression of FOXM1. Knocking down ALKBH5 inhibited FOXM1 expression in GSCs, and restoring FOXM1 could counteract the consequences of inhibiting ALKBH5 on cell proliferation and tumor formation [21]. Together, this study demonstrated that ALKBH5 plays a role in the modulation of cell proliferation and tumorigenesis of GSCs, exerting its influence in part through the regulation of FOXM1.

M^6^A “readers” including members of the YTH domain family exhibit a specificity for recognizing and binding to m^6^A-modified mRNA. They are involved in processes such as RNA splicing, mRNA stability, and translation [6,7,8,9]. Among the YTH domain family, YTHDF2 has been found to be important to normal brain development because YTHDF2 knockout causes the inability of neural stem/progenitor cells to undergo proper proliferation and differentiation [35,36]. Therefore, it is not surprising to find that YTHDF2 plays roles in cell proliferation of glioma cells. The expression of YTHDF2 is elevated in glioblastomas and is associated with an unfavorable patient survival [22]. Knockdown of YTHDF2 in glioblastoma and GSC cells impeded both cell proliferation and DNA replication, suggesting that YTHDF2 protein is essential for glioblastoma cell proliferation. It is of interest that this study also found liver X receptor A (LXRA) to be a target of YTHDF2. LXRA is a major nuclear receptor in maintaining cholesterol homeostasis and has implications for the proliferation of glioblastoma cells. YTHDF2 was found to inhibit LXRA mRNA expression by facilitating its m^6^A-dependent mRNA decay [22]. In another study, YTHDF2 was found to increase UBX domain protein 1 (UBXN1) mRNA degradation in glioblastoma cells [24]. The study also confirmed that the survival outcomes of glioblastoma patients correlate with the expression levels of YTHDF2. Knockdown of YTHDF2 markedly elevated the expression of UBXN1 in the cells, while YTHDF2 overexpression did the opposite. This action of YTHDF2 is dependent on METTL3-mediated m^6^A in gliomas. Additionally, inhibition of UBXN1 by YTHDF2 induced NFκB activation, which promoted cell proliferation and tumor growth [24].

IGF2BP3 (insulin-like growth factor 2 mRNA-binding protein 3) is another m^6^A reader. It is a member of the IGF2 mRNA-binding protein family and is involved in the stabilization and translational control of target mRNAs [37]. A recent study evaluated the impact of IGF2BP3 on the proliferation of glioblastoma cells by using IGF2BP3 shRNAs. Their findings indicate that silencing IGF2BP3 significantly suppressed cell proliferation compared to the control group [26]. Consequently, the knockdown of IGF2BP3 effectively restrained the tumorigenic properties of glioblastoma in a mouse model. Moreover, analysis of TCGA data revealed strong connections between IGF2BP3 and cell cycle regulators, particularly CDK1. As a member of the CDK family, CDK1 plays a crucial role in influencing the cell cycle. The study observed a decrease in CDK1 expression as well as an arrest of glioblastoma cells in the G0/G1 phase upon IGF2BP3 knockdown, underscoring the vital role of IGF2BP3 in cell cycle regulation [26].

### 3.2. Regulating Cancer Cell Stemness

The malignant phenotype of human glioblastoma could be attributed to stem-like cells derived from glioblastoma, known as GSCs. These GSCs serve as the initiators of cancer and exhibit resistance to both chemotherapy and radiation [29,30]. The concept of cancer stem cells was first established by isolation of a minority cell population with stem cell properties; therefore, it was termed stem-like cells [38,39]. GSCs have some similar properties with normal neural stem cells (NSCs), such as capacity for self-renewal. However, some abnormalities have been described in GSCs, including increased proliferation kinetics and abnormal differentiation, with frequent co-expression of markers normally found in separate cell lineages, e.g., neuronal and glial [40,41]. GSCs may have intrinsic properties to avoid differentiating signals from the environment [40,41]. Moreover, GSCs harbor cancer-related molecular abnormalities such as mutation and amplification of the EGFR gene. One of the key issues for our understanding of cancer stem cell biology is defining the molecular circuitry that drives the development and self-renewal in the cancer stem cell.

The vast of majority m^6^A research on glioma was conducted using GSCs due to the above-described importance of GSC in the disease. The role of m^6^A in cancer stem cells was first evidenced when ALKBH5 was found predominately in the niches of cancer stem cells in human glioblastoma tissues [21]. ALKBH5 levels were also higher in GSCs as compared to their matching bulk tumors [21]. Although ALKBH5 is not GSC specific, knockdown of ALKBH5 in GSCs impairs their self-renewal and other stemness properties [21]. ALKBH5 regulates GSC self-renewal largely due to its function on cell proliferation, because cell proliferation is required for GSC self-renewal.

The characters of METTL3 in cancer stem cells of glioblastomas are somewhat contradictory in different publications. Early on, a study demonstrated that METTL3 or METTL14 might be a tumor suppressor for GSCs since knockdown of METTL3 or METTL14 expression by shRNA increased GSC growth and self-renewal [42]. The presence of cell markers associated with differentiation, such as GFAP (astrocyte marker) and Tuj1 (neuronal marker), was also reduced in GSCs with knockdown of METTL3 or METTL14. Conversely, decreased levels of stem cell marker CD44 were noted in GSCs overexpressing METTL3. Moreover, inhibition of METTL3 or METTL14 expression also enhanced tumor growth upon transplantation of GSC cells into mouse brains. To further substantiate the tumor suppressor role of METTL3 or METTL14 n GSCs, changes in RNA transcripts in METTL3- or METTL14-depleted cells were identified for comparison with control cells. Among the changes, oncogenes such as ADAM19, EPHA3, and KLF4 showed increased expression, but tumor suppressors like CDKN2A and BRCA2 demonstrated decreased expression by inhibition of METTL3- or METTL14. The changes in these down-stream target genes further support the claim that METTL3 and METTL14 have tumor suppressor functions in GSCs [42].

In contrast, a study from a different research group indicated that METTL3 has oncogenic properties on GSCs. The results in the report showed that high levels of METTL3 expression have been observed in glioma tissues compared to normal brain tissues, and METTLE3 expression informed worse survival of glioma patients [20]. By using GSCs and established glioma cell lines, the authors found that METTL3 is essential for maintaining GSCs and reprogramming of the established glioma cell lines [20]. Silencing the METTL3 gene by shRNA led to a decrease in self-renewal, determined by neurosphere formation assay and limiting dilution assay as well as in the expression of the stem cell-specific marker Stage-Specific Embryonic Antigen 1 (SSEA1). The expression of SSEA1 is often used as a characteristic feature of undifferentiated or pluripotent stem cells. Thus, the results suggest a potential role of METTL3 in stem cell maintenance or differentiation processes. In addition, METTL3 silencing in GSC cells METTL3 resulted in reduced expression of glioma reprogramming factors, including SOX2 (SRY-box 2) [20]. SOX2 is a transcription factor that plays a crucial role in the regulation of embryonic development and maintenance of stem cell pluripotency. SOX2 is known to maintain stem cell properties in normal neural stem cells. In the context of glioma, the expression of SOX2 is associated with the preservation of stemness of GSCs. SOX2 has been implicated in various aspects of tumor initiation and maintenance. Moreover, high levels of SOX2 expression in glioma cells are often correlated with increased tumor grade and aggressiveness. In connection with m^6^A RNA modification, SOX2 was shown to be a direct target of METTL3 [13]. METTL3 protein interacted with SOX2 mRNA and stabilized SOX2 mRNA. METTL3 silencing results in a decreased level in SOX2 mRNA in GSC cells. The effect of METTL3 silencing on neurosphere formation of GSC cells was rescued by re-expression of SOX2 [20]. Together, these data suggested that SOX2 could be an important factor that executes the oncogenic function of METTL3 in GSC.

Another independent study examining the role of METTL3 in GSCs using CRISPR/Cas9 gene targeting to knock out METTL3 reported that METTL3 is indispensable for GSC growth and maintenance of stemness. Knockout of METTL3 in GSC cells suppressed GSC proliferation and stemness [19]. In this report, the mechanism of the roles of METTL3 in GSCs was linked to mitophagy [19]. Mitophagy is a process that removes damaged or dysfunctional mitochondria, thereby ensuring that the cellular mitochondrial population is in a healthy state and functioning properly. The loss of mitophagy function may disrupt these protective mechanisms and potentially contribute to tumorigenesis. Knockout of METTL3 in GSC cells activated mitophagy process cells, whereas overexpression of METTL3 in the cells suppressed mitophagy [19]. Moreover, METTL3 formed a complex with METTL14, which was also involved in the regulation of mitophagy in GSC cells through methylation of OPTN (optineurin) mRNA, which is known to be involved mitophagy. These data further support the oncogenic roles of METTL3 in GSCs.

Collectively, findings regarding the roles of METTL3 in glioma across different studies are inconsistent. At present, there is no clear and undisputed explanation for these contradictory results. Consequently, caution is advised when considering METTL3 as a therapeutic target for glioma. This uncertainty could arise from various factors, including differences in study methodologies, cell lines, or even variations in the genetic makeup of the gliomas being studied. Further investigation with more standardized protocols or larger sample sizes is needed to reconcile these discrepancies and provide a more comprehensive understanding of METTL3’s involvement in glioma.

### 3.3. Regulating of Cell Apoptosis and Other Types of Cell Death

Apoptosis, often referred to as programmed cell death, plays a crucial role in maintaining tissue homeostasis, and cancer cells often evade or resist this process. Cancer cells have the ability to proliferate uncontrollably and avoid cell death mechanisms, which results in the development and progression of tumors. METTL3’s oncogenic role in glioblastoma is further manifested through its modulation of apoptotic signaling pathways [20]. In 2019, a new study validated earlier discoveries by demonstrating elevated METTL3 and YTHDF2 levels in samples from glioblastoma patients as compared with normal brain tissues [43]. Likewise, the researchers confirmed that silencing METTL3 through knockdown was shown to inhibit tumor growth in mouse models. Moreover, their RNA sequencing results revealed an enrichment of carcinogenesis-related pathways, such as apoptotic signaling pathways, among the set of m^6^A-regulated genes in METTL3-silenced glioblastoma cell lines [43]. The changes in apoptotic signaling pathways involved BCL-X splicing variants (both anti-apoptotic and pro-apoptotic proteins) that are intrinsic regulators of apoptosis. Specifically, the upregulation of the pro-apoptotic variant BCL-XS was observed in METTL3-silenced cells, while the anti-apoptotic variant BCL-XL showed increased expression in control glioblastoma cells [43]. This alteration was due to METTL3 silencing causing a pronounced decrease in the transcription of BCL-XL, favoring the generation of BCL-XS transcript. Furthermore, the authors investigated the impact of BCL-X splicing alteration on METTL3-silencing phenotypes by combined knockdown of METTL3 and BCL-XS in glioblastoma cells. They found that combined knockdown of METTL3 and BCL-XS resulted in a notably accelerated growth rate and decreased apoptosis compared to cells subjected to METTL3-silencing alone [43]. This study also reveals another novel discovery: YTHDC1, a reader of m^6^A modification, was required for the METTL3-mediated alteration of alternative splicing. Therefore, METTL3 couples with YTHDC1 to play a role in the anti-apoptosis of glioblastoma cells, partially mediated through the splicing modulation of BCL-X.

Ferroptosis is another form of regulated cell death that is characterized by the iron-dependent accumulation of lipid peroxides to lethal levels. The process of ferroptosis is associated with the disruption of cellular redox balance and the accumulation of lipid peroxides, leading to cell damage and death. Cancer cells, including glioblastoma cells, often exhibit alterations in their redox status and are sensitive to oxidative stress. Targeting the pathways involved in ferroptosis could be a strategy to selectively eliminate cancer cells. Recently, ALKBH5 has been shown to regulate ferroptosis through GCLM (Glutamate-Cysteine Ligase Modifier Subunit) [44]. GCLM plays a crucial role in the production of glutathione, serving as a vital antioxidant that safeguards cells against oxidative harm—a phenomenon linked to ferroptosis. The ALKBH5–GCLM axis suppressed ferroptosis and promoted tumor cell viability, thereby increasing survival of the mice bearing glioblastoma. However, the roles of ALKBH5 in glioblastoma stemness are independent from GCLM, because modulating GCLM expression using shRNAs or an inhibitor did not significantly impact the levels of GSC stemness markers or self-renewal [44]. This implies that other targets of ALKBH5, rather than GCLM, regulate stemness. Moreover, knockdown of ALKBH5 in GSCs resulted in accelerated degradation of GCLM transcripts. YTHDF2 was identified as a binding partner for GCLM transcripts, and the knockdown of ALKBH5 expression enhanced the association between YTHDF2 and GCLM mRNA. Blocking YTHDF2 function effectively reversed the alterations in GCLM levels induced by ALKBH5 knockdown. These findings suggest that ALKBH5 upregulates GCLM mRNA levels by impeding YTHDF2-mediated decay.

### 3.4. Regulating Cell Migration and Invasion

The ability of glioma cells to migrate and invade normal brain tissue highlights a significant challenge in treating gliomas. Treatment for gliomas typically involves surgery, which aims to remove as much of the tumor as possible without causing damage to essential brain functions. However, gliomas often infiltrate surrounding brain tissue, making complete surgical removal difficult and leading to eventual tumor recurrence at the edges of the treated areas. Therefore, understanding the invasion mechanisms is crucial for developing targeted therapies to inhibit glioma progression and recurrence.

It has been shown that YTHDF2 regulates invasiveness of GSCs and glioblastoma cells [22]. Knockdown of YTHDF2 in GSC cells led to the suppression of in vitro cell invasiveness, whereas the overexpression of YTHDF2 in glioblastoma cells increased their invasion in vitro. Moreover, in vivo, YTHDF2-depleted tumors exhibited distinct margins with significantly reduced invasive tumor areas. The reintroduction of a shYTHDF2-resistant YTHDF2 into YTHDF2-depleted GSCs rescued the malignant phenotypes of GSCs in vitro and in vivo. Furthermore, a downstream target of YTHDF–LXRA has been identified as a regulator of glioblastoma cell invasion. YTHDF2 interacted with LXRA mRNA, resulting in the upregulated decay of LXRA mRNA and thereby promoting cell invasion of GSCs and glioblastoma cells [22].

IGF2BP2 exhibits elevated expression in glioblastoma, where it plays a role in governing the migratory and invasive capabilities of the cells. Its mechanism involves the downregulation of E-cadherin and the facilitation of increased expression of Vimentin and N-cadherin [45]. Reduced expression of E-cadherin is often associated with increased invasiveness and metastasis in various cancers. Similar to E-cadherin, alterations in N-cadherin expression can contribute to cancer progression. Increased N-cadherin expression is associated with enhanced cell motility and invasion. Vimentin is a type III intermediate filament protein that is a component of the cytoskeleton. Cancer cells with elevated vimentin levels may display enhanced migratory and invasive properties. Collectively, IGF2BP2 in glioblastoma is linked to the regulation of cell migration and invasion through the modulation of these key proteins. Additionally, another member of the IGF2BP family, IGF2BP3, is similarly upregulated in glioma tissues compared to normal brain tissues, contributing to the invasive potential of glioma cells [45].

### 3.5. Regulating Tumor Immunity

Investigation of the participation of m^6^A regulators in glioblastoma and their connection with the tumor immune microenvironment (TIME) has been reported [46]. In a sentinel study, a comprehensive collection of potential m^6^A RNA regulators was acquired and an evaluation of PD-L1 and PD-1 levels, immune cell infiltration, and immune scores was conducted. The authors found that the expression levels of most m^6^A regulators including METTL3, METTL14, WTAP, YTHDF2, YTHDF3, YTHDF1, RBM15, FTO, and ALKBH5 are high in glioblastoma tissues [46]. Also, PD-L1 and PD-1 exhibited significant upregulation in glioblastoma tissues. Some of regulators demonstrated a notable augmentation in immune score, along with increased levels of monocytes, M1 macrophages, activated mast cells, and eosinophils [47]. For example, ALKBH5 displayed significant associations with TIME and manifested positive correlations with PD-L1. The findings suggest that m^6^A methylation regulators likely play a pivotal role in modulating PD-L1 expression and immune infiltration, exerting a substantial influence on the glioblastoma TIME.

The impact of ALKBH5 expression on the glioma tumor immune microenvironment has been explored by another study. Findings indicated high ALKBH5 expression correlated with elevated scores for genes associated with tumor immunity such as lymphocyte-specific kinase (LCK), major histocompatibility complex I (MHC-I), major histocompatibility complex II (MHC-II), and signal transducer and activator of transcription 1 (STAT1) in glioma patients [47]. This implies that ALKBH5 expression may be involved in governing interferon signaling, lymphocyte activation, and activation of antigen-presenting cells within gliomas.

ALKBH5 has been demonstrated to be necessary in facilitating the infiltration of tumor-associated microglia or macrophages (TAM) in vivo under hypoxic conditions. These hypoxia-induced TAMs exhibit M2-like macrophage characteristics associated with immunosuppressive functions. Tumors lacking active ALKBH5 or those depleted of ALKBH5 exhibited a notably reduced percentage of TAMs compared to control groups, indicating the essential role of ALKBH5’s demethylase activity in TAM recruitment and immunosuppression [48]. Additionally, the key cytokine gene CXCL8/IL8, crucial for TAM recruitment and immunosuppression, is dependent on ALKBH5 activity [38]. Downregulation of ALKBH5 in glioblastoma cells results in a significant decrease in CXCL8/IL8 production, leading to diminished recruitment of hypoxia-induced TAMs and attenuated immunosuppression. Furthermore, CXCL8 expression restored TAM abundance in ALKBH5-deficient tumors [48]. In summary, the findings suggest that CXCL8 expression plays a crucial role, at least in part, in mediating ALKBH5-induced TAM recruitment and immunosuppression.

PD-L1 can bind to PD-1 and inhibits the activity of immune cells. Some cancer cells can express PD-L1 to evade detection by the immune system. ALKBH5 has been documented to facilitate PD-L1-mediated immune evasion in glioma [49]. Depletion of ALKBH5 resulted in heightened T cell infiltration within gliomas. Specifically, ALKBH5 knock out disrupted the YTHDF2-mediated stability of ZDHHC3 mRNA, consequently hindering PD-L1 expression by expediting PD-L1 degradation. The diminished PD-L1 protein levels associated with ALKBH5 deficiency were attributed to the suppression of ZDHHC3 mRNA expression in an m^6^A modification-dependent manner. Reducing ZDHHC3 inhibited PD-L1 expression by hastening the degradation of PD-L. Furthermore, targeting ALKBH5 was found to enhance the tumor immune microenvironment and boost the effectiveness of anti-PD-1 therapy by accelerating PD-L1 degradation [49]. Table 1 summarizes the functions and downstream target genes of m^6^A modulators in glioma discussed in this review.

## 4. Mechanisms for Dysregulation of m^6^A-Related Genes

The development and progression of gliomas involve complex signaling pathways. Several molecular and genetic alterations contribute to the initiation and growth of gliomas including p53, EGFR and PDGFR pathways. Studies have shown that these pathways facilitate upstream regulation of m^6^A in glioblastoma. Understanding the connections between these pathways and the dysregulation of m^6^A-related genes would enhance our comprehension of the molecular mechanisms underlying glioma formation.

Investigations were conducted to unravel the process behind the elevated expression of YTHDF2 in glioblastoma. The findings suggest that the activation of the epidermal growth factor receptor (EGFR) contributes to YTHDF2 overexpression in glioblastoma, thereby linking EGFR signaling, a pivotal factor in the progression and resistance to therapy of glioma, to m^6^A-dependent mRNA modification [22]. Specifically, the overexpression of YTHDF2 in glioma is due to the activation of EGFR/SRC/ERK pathway [22]. Notably, the stabilization of YTHDF2 protein was ensued through the phosphorylation of serine39 and threonine381 sites on the protein by the extracellular regulated MAP kinase (ERK). Moreover, the EGFR–YTHDF2 axis emerges as a pivotal mechanism that governs the downregulation of LXRA gene expression, playing a crucial role in cholesterol dysregulation within the context of glioblastoma tumorigenesis. Importantly, the YTHDF2 protein expression in GSC cells could be suppressed by kinase inhibitors targeting EGFR, SRC, and ERK1/2, implying the therapeutic potential of these inhibitors on YTHDF2.

EGFR signaling can also suppress m^6^A levels in glioblastoma by inhibiting ALKBH5 nuclear export, thereby enhancing the function of the m^6^A eraser [44]. The nuclear localization of ALKBH5 is important to its function because RNA m^6^A demethylation often occurs in the nucleus. Treatment with EGF to activate EGFR signaling pathway led to the translocation of ALKBH5 from the cytoplasm to the nucleus in GSC cells. The expression of the constitutively active EGFR variant, EGFRvIII, facilitated the nuclear localization of ALKBH5 in GSC cells. Conversely, silencing EGFR expression in GSC cells using shRNAs impeded the nuclear localization of ALKBH5. Similarly, the administration of the EGFR inhibitor erlotinib resulted in increased localization of ALKBH5 in the cytoplasm. Overexpression of EGFR and EGFRvIII triggered the phosphorylation of ALKBH5 through SRC, while the use of EGFR inhibitors diminished the phosphorylation of ALKBH5 in GSC cells. Additionally, the phosphorylation of ALKBH5 at tyrosine 71 by SRC was identified as a crucial event for ALKBH5 nuclear localization. Furthermore, Exportin, also known as chromosomal maintenance 1 (CRM1), was found to mediate the nuclear export of ALKBH5 in GSC cells. EGF/SRC-induced phosphorylation of ALKBH5 resulted in the inhibition of the binding between ALKBH5 and CRM1, thereby reducing the transport of ALKBH5 from the nuclei to the cytoplasm [44].

PDGFR signaling is another important oncogenic pathway in glioma. In contrast to EGFR, PDGFR signaling upregulates m^6^A levels in glioblastoma [19]. The mechanism for this action of PDGFR is facilitated by regulating METTL3 through transcriptional control [19]. Overexpression of PDGFRa or PDGFRb in GSC cells led to an increase in m^6^A levels, with suppression of PDGFRb expression using shRNAs or PDGFR inhibitors resulted in a reduction in the overall m^6^A levels in GSC cells. This implies that PDGFR signaling is involved in upregulating m^6^A in GSCs. Moreover, adding PDGF to GSC cells resulted in the upregulation of METTL3 mRNA and an increase in METTL3 protein levels. In contrast, knockdown of PDGFRb led to a reduction in METTL3 mRNA and protein levels. PDGF enhances the transcription of METTL3 through Early Growth Response 1 (EGR1) in GSC cells. EGR1 is a downstream target of PDGF signaling. The expression of METTL3 shows a positive correlation with EGR1 expression in human glioma tissues. Binding of EGR1 to the METTL3 promoter in GSC cells is indicative of its regulatory role. Elevated levels of EGR1 result in increased activity of the METTL3 promoter. Furthermore, the enforced expression of EGR1 successfully restores METTL3 expression that was diminished in the PDGFRb knockdown GSC cells. Collectively, this study elaborated a novel function of PDGF, wherein it governs the transcription of METTL3, leading to consequential changes in m^6^A levels in glioma.

Recently, we reported a mechanism centered on the posttranslational modification of ALKBH5. In this process, the deubiquitinase USP36 plays a pivotal role in enhancing the protein stability of ALKBH5 by directly removing ubiquitin molecules from the protein [53]. Initially, we found that the ubiquitin-proteasome proteolytic pathway is responsible for the degradation of ALKBH5 protein. Given that ALKBH5 protein is quite stable in glioma cells, we focused on identifying deubiquitinases capable of impeding the ubiquitination-mediated degradation of ALKBH5. Through screening 42 deubiquitinases, 8 promising candidates were identified. Among these candidates, USP36 emerged as the most effective. Subsequent mass spectrometry experiments were conducted to unveil the protein interaction partners of ALKBH5, and the results demonstrated the interaction between USP36 and ALKBH5 [53]. Moreover, USP36 and ALKBH5 proteins co-localized and exhibited a direct interaction in GSC cells. Knockout of USP36 in GSC cells resulted in a notable accumulation of ubiquitinated ALKBH5. Consequently, the deletion of USP36 significantly increased the degradation of ALKBH5, while overexpression of USP36 enhanced ALKBH5 stability in GSC cells. Additionally, in vitro deubiquitination assays with purified ALKBH5, HA-Ubi, and USP36 proteins demonstrated that USP36 directly deubiquitinated ALKBH5 protein. These findings solidify USP36 as the deubiquitylating enzyme crucial for maintaining the stability of ALKBH5 [53].

The clinical significance of the USP36-ALKBH5 pathway was assessed by investigating the expression of the USP36 protein in both normal human brain tissues and gliomas. In gliomas, the levels of USP36 protein were found to be elevated compared to normal brain tissues, and this elevation exhibited a positive correlation with the malignant grade of gliomas [53]. Importantly, the expression levels of ALKBH5 and USP36 protein correlated with each other in the human gliomas [54]. Furthermore, in GSC cells, knockout of USP36 inhibited the cell proliferation, stemness and tumorigenicity. The reintroduction of ALKBH5 into the USP36-deleted GSC cells significantly mitigated the inhibitory impact observed with USP36 knockout, particularly in cell proliferation, stemness, and tumor growth of the GSC cells. Together, these findings underscore the crucial role of the USP36–ALKBH5 axis in governing GSC cell proliferation, stemness and tumorigenesis, and could potentially lay the groundwork for progressing the development of more potent therapies targeting glioblastoma [53,54]. Figure 1 illustrates the various regulators and mechanisms for dysregulation of m^6^A-related genes discussed in this sector.

## 5. The Therapeutic Potential and Treatment Resistance Roles of m^6^A Modulators

Exploring the treatment resistance and therapeutic possibilities associated with m^6^A-related proteins is an evolving focus of research, particularly in the field of cancer. The roles of m^6^A-related proteins in treatment resistance are complex and context-dependent. Understanding the mechanisms behind these proteins and how small molecules can modulate them is crucial for improving the effectiveness of existing cancer treatments.

### 5.1. Treatment Resistance

Studies have implicated m^6^A modifications in the development of resistance to chemotherapeutic agents including TMZ in glioma. Dysregulation of m^6^A writers, erasers, or readers can impact the expression of genes involved in drug response and resistance. For example, the USP36–ALKBH5 axis has been shown to impart resistance to TMZ, the first-line chemotherapeutic agent for standard practice in the management of glioblastoma [53]. ALKBH5 is essential for the stemness of GSCs that confers resistance to TMZ. Accordingly, knockout of the USP36 gene downregulated ALKBH5 expression, thereby significantly increasing the sensitivity of GSC cells to TMZ treatment [54]. Similarly, another study demonstrated that the demethylation of m^6^A by ALKBH5 has the potential to enhance the levels of stem cell transcription factors NANOG and SOX2, leading to the promotion of drug resistance in gliomas [55].

METTL3 also plays a crucial role in sustaining resistance to TMZ in glioblastoma cells [50]. Knockdown of METTL3 increased the sensitivity of TMZ-resistant glioblastoma cells to TMZ treatment, while overexpression of METTL3 heightened resistance to TMZ. Moreover, knocking down METTL3 significantly decreased the self-renewal capacity of TMZ-resistant cells but increased apoptosis following TMZ treatment in the cells. In in vivo experiments, METTL3 knockdown in glioblastoma cells resulted in a significant reduction in tumor growth and enhanced TMZ sensitivity compared to control cells. Mice injected with shMETTL3 glioblastoma cells exhibited improved survival rates and greater responsiveness to TMZ therapy compared to the mice injected with control cells. Mechanistically, METTL3 regulated the expression of EZH2 that contributed to increased resistance to TMZ in glioblastoma cells [50]. These findings strongly support the role of METTL3 in TMZ resistance in glioblastoma cells.

The radioresistance of cancer cells poses a significant challenge in the treatment of glioblastoma because this radioresistance can contribute to local recurrence of the cancer. A potentially effective strategy could involve focusing on the factors responsible for radioresistance, aiming to enhance the effectiveness of radiotherapy. ALKBH5 has been shown to be important to radioresistance of glioblastoma [51]. Depletion of ALKBH5 expression in glioblastoma stem cells diminished the survival of the cells following irradiation when compared to control cells, suggesting that downregulation of ALKBH5 enhances radioresistance. In cells with ALKBH5 depletion, the lower survival of glioblastoma stem cells post-irradiation is likely attributable to a reduction in the expression of genes involved in damage response, such as CHK1 and RAD51 [51]. Silenced METTL3 in GSC cells also exhibited increased responsiveness to γ-irradiation and impaired DNA repair, as indicated by the significant accumulation of γ-H2AX [13]. Consequently, targeting ALKBH5 or METTL3 may emerge as a promising therapeutic strategy to counteract the radioresistance observed in glioblastoma.

### 5.2. Therapeutic Potential of Small Molecules Targeting m^6^A Modulators

Development of small molecule activators or inhibitors that target m^6^A writers, erasers, or readers is underway. Considering the significant roles of m^6^A modifications in the development and progression of cancer, targeting writers, erasers, or readers could be a promising therapeutic strategy. Researchers are currently investigating small molecule inhibitors that can modulate these proteins for their potential in the treatment of glioma.

The therapeutic effectiveness of UZH1, a METTL3 inhibitor, was evaluated in GSC cells through in vitro testing and in an animal model [12]. In vitro experiments illustrated UZH1’s efficacy in reducing the viability of GSCs, while in vivo studies demonstrated its potency in suppressing tumor growth. FTO inhibitors have also been investigated for their therapeutic potential in glioblastoma. Treatment using MA2, an inhibitor for FTO, significantly inhibited the proliferation of GSC cells in vitro [33]. Moreover, administering MA2 resulted in a remarkable suppression of GSC-induced tumorigenesis and prolonged the survival of animals bearing GSC tumors [33].

Novel FTO inhibitors have been crafted and synthesized through molecular docking studies of FTO, exhibiting low micromolar IC50 values and a distinct preference for FTO over ALKBH5 [45]. Among these inhibitors, two competitive compounds have emerged as leading candidates: FTO-02 and FTO-04. At a concentration of 30 μM, these inhibitors demonstrated a remarkable reduction in neurosphere formation of GSC cells. Notably, FTO-04 exhibited a significant ability to disrupt the self-renewal of GSCs but did not adversely affect the growth of human normal neural stem cell neurospheres. Crucially, treatment with FTO-04 led to an elevation in m^6^A methylation level [52]. Additional research endeavors aimed to enhance FTO-04 by elevating both potency and selectivity through the rational design of oxetanyl-class compounds [56]. This approach led to the identification of several compounds exhibiting nanomolar IC50s against recombinant FTO. The lead compound FTO-43 N demonstrated the inhibitory effects on the growth of glioblastoma cells at nanomolar concentrations [56]. These findings suggest the development of an improved class of FTO inhibitors.

Researchers have discovered novel inhibitors targeting ALKBH5 through high-throughput screening of the Enamine Pharmacological Diversity Set, a collection comprising 10,240 pure compounds [57]. The subsequent assessment of enzyme kinetics inhibition and the anticipated positioning of docking on ALKBH5 resulted in the identification of two inhibitors: Ena15 and Ena21. The effectiveness of both inhibitors was tested in inhibiting the cell proliferation of glioblastoma cells. The results demonstrated that both inhibitors effectively suppressed the proliferation of glioblastoma cells and halted the progression of the cell cycle at the G0/G1 phase. Additionally, when glioblastoma cells were treated with these inhibitors, there was an observed elevation in m^6^A levels. The observations suggest Ena15 and Ena21 are compounds with a distinct ability to selectively inhibit the activity of ALKBH5 in glioblastoma cells [57].

Recently, a brand-new CRISPR-based technology, dCasRx, has been used to precisely edit the mRNAs of METTL3 and ALKBH5 in glioblastoma cells [58]. “dCasRx” refers to a modified version of the CRISPR-Cas gene-editing system in which the Cas enzyme is ‘dead’ or inactivated. Specifically, dCasRx is designed for RNA targeting. It can be used to manipulate gene expression at the RNA level by fusing the dCas protein with RNA-targeting domains, allowing it to bind to specific RNA sequences. The investigation utilized the CRISPR-Cas13 family, known for its ability to bind and cleave single-stranded RNA (ssRNA) guided by complementary guide RNA. Specifically, dCas13Rx, exhibiting significant editing efficiency with minimal off-target activity, was fused to METTL3 or ALKBH5 [58]. Both the dCasRx-METTL3 and dCasRx-ALKBH5 constructs enabled site-specific m^6^A installation. dCasRx-METTL3 expression resulted in modified m^6^A methylation on the FOXM1 transcript and decreased FOXM1 mRNA levels. Conversely, dCasRx-ALKBH5 mediated m^6^A demethylation on the MYC transcript, leading to downregulation of MYC mRNA levels. The subsequent reduction in FOXM1 or MYC mRNA induced by these dCasRx edits ultimately inhibited the proliferation of GSC cells [58]. Together, these discoveries suggest that the dCasRx system can be a powerful tool in investigating formerly unclear site-specific m^6^A modifications in RNAs and clarifying the causal relationships between m^6^A modifications and phenotypes. Furthermore, the compact size of the dCasRx epitranscriptomic editors enables efficient packaging in lentiviruses, making them viable for therapeutic applications [58].

## 6. Conclusions

The impact of m^6^A mRNA methylation extends broadly, particularly in its contributions to normal cellular functions and the initiation and progression of cancers. Its various roles often involve the intricate interplay of distinct signaling cascades. It is crucial to emphasize that the field continues to evolve, and additional research is essential to gain a comprehensive understanding of the molecular mechanisms underlying the intricate interplay among M^6^A mRNA modification and cellular processes. Moreover, future investigations should prioritize collaborative efforts to navigate the complexities of patient data analysis, interpretation, and application. The aim is to enhance our comprehension of m^6^A RNA modification and its implications for advancing targeted therapies and immunotherapy as well as for optimizing drug delivery. Additionally, delving deeper into the potential therapeutic advantages of targeting crucial m^6^A RNA methylation modulators that drive malignant dysregulations in glioma represents another key objective for future research. Furthermore, the toxicity and adverse effects of inhibitors targeting m^6^A modulators must undergo comprehensive evaluation in animal models, followed by rigorous assessment in clinical trials. It is crucial to carefully weigh the potential benefits against the associated risks to ensure patient safety. Given the vital functions of m^6^A modulators in controlling multiple aspects of glioma, managing the levels of these regulators could potentially be a viable strategy for overcoming the therapeutic resistances of the disease. Ongoing research and advancements in this field inspire hope for improved treatments and outcomes for glioma in the future.

## Figures and Tables

**Figure 1 cancers-16-00727-f001:**
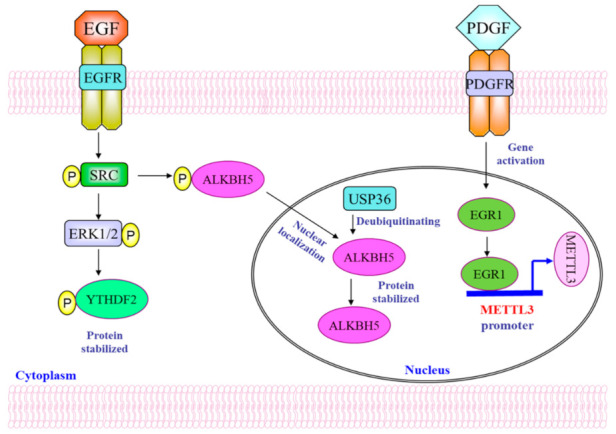
The figure demonstrates the wide array of mechanisms present in the dysregulation of m^6^A-related genes in gliomas. P: phosphorylation.

**Table 1 cancers-16-00727-t001:** Functions and downstream target genes of m^6^A modulators in glioma.

Gene Name	Functions	Downstream Tagets	References
METTL3	Maintaining GSC stemness	Sox2, SSEA1	[20]
	Suppression of GSC stemness	ADAM19, EPHA3, KLF4, CDKN2A	[42]
	Suppression of mitophagy	OPTN	[19]
	anti-apoptosis	BCL-X	[43]
	TMZ resistance	EZH2	[50]
	Radioresistance	γ-H2AX	[20]
ALKBH5	Cell-cycle, Maintaining GSC stemness	FOXM1	[21]
	Suppression for Ferroptosis	GCLM	[44]
	Hypoxia-induced TAM	CXCL8	[48]
	PD-L1-mediated immune evasion	ZDHHC3	[49]
	TMZ resistance	NANOG, SOX2	[50]
	Radioresistance	CHK1, RAD51	[51]
FTO	Maintaining GSC stemness	ADAM19, EPHA3, KLF4,	[42]
	Tumor growth		[52]
YTHDF2	Cell-proliferation	LXRA	[22]
	Cell-proliferation	UBXN1	[24]
	Invasion	LXRA	[22]
YTHDC1	anti-apoptosis	BCL-X	[43]
IGFBP2	Migration and invasion	Vimentin, N-cadherin, E-cadeherin	[45]
IGF2BP3	Cell proliferation	CDK1	[26]

## Data Availability

The data can be shared up on request.

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
