# Peer review of "m6A mRNA Modifications in Glioblastoma: Emerging Prognostic Biomarkers and Therapeutic Targets"

_cancers, 2024, doi:10.3390/cancers16040727_

Round 1

Reviewer 1 Report

Comments and Suggestions for Authors

This review article entitled “m6A mRNA Modifications in Glioblastoma: Emerging Prognostic Biomarkers and Therapeutic Targets” introduces an emerging important and new area in Epitranscriptomics by focusing on m6 mRNA modification in the glioblastoma, a very aggressive type of cancer. The authors have provided a comprehensive overview of progress in this relative new field, and discuss underlying mechanisms and therapeutic potential and challenges. It is well-written, which will attract wide interest of readers in the field of not only cancer research but also biomedical studies in general. It can be accepted at present form.

Author Response

We thanks the reviewer for the positive comments.

Reviewer 2 Report

Comments and Suggestions for Authors

In their manuscript, Xie et al offer a comprehensive exploration of m6A RNA modification in glioma, covering the prognosis of m6A-related genes, functions and downstream target genes, upstream regulatory mechanisms, and the therapeutic potential of m6A modulators. Overall, the manuscript is well-written with a clear presentation, and the references are accurately cited. Consequently, it holds potential for publication in Cancers. The authors are encouraged to address the following suggestions to further enhance the manuscript:

1.     Consider providing a brief overview of the development of techniques used to profile m6A modification. Additionally, in section three, "m6A RNA methylation in gliomas: functions and downstream target genes," specify which targets are directly regulated by m6A modification on its transcript.

2.     Add citations for each downstream target in Table 1. Correct the column names to read "Downstream targets" and remove the second empty row.

3.     The  last paragraph of section 5 about dCasRx system is inaccurate. First, Cas13 system recognize and cut RNA but not DNA. Mutation in dCas13 prevent it from cutting “RNA” but not “DNA”. Moreover, in line 615, “dCas13Rx-mediated targeting of METTL3 and ALKBH5” in not accurate. It is fused with METTL3 or ALKBH6 protein or domains, rather than targeting these two genes.

4.     Consider adding a table to summarize the content of section 2 about prognosis.

5.     The detailed introduction of the TCGA project from line 77-86 seems unnecessary.

6.     In line 166, modify "brain tumor rate" to "brain tumor formation rate" and change "small" to "smaller" for improved clarity.

Comments on the Quality of English Language

The quality of English language is acceptable. A few spelling errors need to be revised.

Author Response

Please seethe attached file

Reviewer 3 Report

Comments and Suggestions for Authors

The paper entitled " " by Xie et al provides insights into glioblastoma, the most common malignant CNS tumor in the United States, emphasizing its poor prognosis and limited treatment success. It delves into the role of N6-methyladenosine (m6A) RNA modification in regulating gene expression and its impact on glioma prognosis. Various m6A regulators, such as METTL3, ALKBH5, YTHDF2, and IGF2BP3, are discussed, with elevated expression levels associated with poor prognosis. The text also explores the establishment of an m6A regulatory gene signature as a potential biomarker for predicting glioma outcomes. Additionally, downstream target genes influenced by m6A RNA modification are highlighted, particularly in cell proliferation, cell cycle, stemness, immunity, and invasion. The paper also talks about therapeutic potential drugs that target m6A modifiers.

The author does a good job in breaking down all the things necessary to understand the big picture of the review as well in highlighting the role of m6A in glioblastoma. I would like the reviewer to add details about how m6A modifications can affect binding sites for RNA Binding proteins and micro RNAs leading to changes in mRNA stability especially for YTHDF. The references are as follows:

Alarcon CR, Lee H, Goodarzi H, Halberg N, Tavazoie SF. N6-methyladenosine marks primary microRNAs for processing. Nature. 2015;519(7544):482–5.

Xu C, Yuan B, He T, Ding B, Li S. Prognostic values of YTHDF1 regulated negatively by mir-3436 in glioma. J Cell Mol Med. 2020;24(13):7538–49.

Comments on the Quality of English Language

Nothing!

Author Response

Please see the attached file,

Reviewer 4 Report

Comments and Suggestions for Authors

The article focusses on the so-called N6-methyladenosine modification of mRNA, which is a rapidly evolving area of research. N6-methyladenosine (m6A) modification plays a pivotal role in post-transcriptionally regulating gene expression and biological functions in cancer as well as in other disorders. The m6A modification is not the only, but one of the most important epigenetic modifications found in glioblastoma. Downstream effects of this modification are described in details as well as the very complex mechanisms of interactions and regulation of gene expression.

The article includes a review of modulators targeting m6A including potential therapeutic perspectives. As epigenetic modifications are basically reversible, respective modulators are in the focus of research. The article is well written and provides an updated review on the m6A modification of mRNA, which ranks among the of the most important epigenetic modifications found in glioblastoma.

Author Response

We thank the reviewer for the positive comments.